# Design, Synthesis and Biological Evaluation of Diosgenin-Amino Acid Derivatives with Dual Functions of Neuroprotection and Angiogenesis

**DOI:** 10.3390/molecules24224025

**Published:** 2019-11-07

**Authors:** Desheng Cai, Jinchai Qi, Yuqin Yang, Wenxi Zhang, Fei Zhou, Xiaohui Jia, Wenbo Guo, Xuemei Huang, Feng Gao, Hongshan Chen, Tong Li, Guoping Li, Penglong Wang, Yuzhong Zhang, Haimin Lei

**Affiliations:** School of Chinese Pharmacy, Beijing University of Chinese Medicine, Beijing 102488, China; 20170931818@bucm.edu.cn (D.C.); 17862969559@163.com (J.Q.); yangyq5@163.com (Y.Y.); zhangwxnn@126.com (W.Z.); zf116318@163.com (F.Z.); 18811508865@163.com (X.J.); wb_guo@126.com (W.G.); hxm3928@163.com (X.H.); gaofeng_1996@126.com (F.G.); chs1314as@163.com (H.C.); lt1755258545@163.com (T.L.); lgp2724760063@163.com (G.L.); wpl581@126.com (P.W.)

**Keywords:** diosgenin-amino acids derivatives, neuroprotection, angiogenesis, CAM model

## Abstract

Diosgenin, a natural product with steroidal structure, has a wide range of clinical applications in China. It also shows great potential in the treatment of blood clots and nerve damage. To enhance the bioavailability as well as efficacy of diosgenin, eighteen diosgenin-amino acid derivatives were designed and synthesized. The neuroprotective effects of these compounds were evaluated by SH-SY5Y cell line and the biosafety was evaluated by H9c2 cell line. The results displayed that part of the derivatives’ activities (EC_50_ < 20 μM) were higher than positive control edaravone (EC_50_ = 21.60 ± 3.04 μM), among which, DG-15 (EC_50_ = 6.86 ± 0.69 μM) exhibited the best neuroprotection. Meanwhile, biosafety evaluation showed that DG-15 had no cytotoxicity on H9c2 cell lines. Interestingly, combined neuroprotective and cytotoxic results, part of the derivatives without their protecting group were superior to compounds with protecting group. Subsequently, Giemsa staining and DAPI (4′,6-diamidino-2-phenylindole) staining indicated that DG-15 had a protective effect on damaged SH-SY5Y cells by reducing apoptosis. Moreover, DG-15 showed a higher role in promoting angiogenesis at high concentrations (4 mg/mL) on the chorioallantoic membrane model. This finding displayed that DG-15 had dual functions of neuroprotection and angiogenesis, which provided further insight into designing agent for the application in treatment of ischemic stroke.

## 1. Introduction

Ischemic stroke remains the third leading cause of mortality and long-term disability worldwide and thousands of people suffer strokes, living with some form of neurological impairment or disabilities each year. Ischemic stroke is related to insufficient blood supply in part or whole of the brain [1,2,3]. Inspired by this pathological occurrence, researchers have leaned towards enhancing the efficiency of angiogenesis [4], dissolving thrombus or salvaging ischemic neurons from irreversible injury [5] as the therapeutic strategy. Currently, many drugs, such as tissue plasminogen activator (tPA), heparin, urokinase, edaravone and nimodipine, were commonly used in clinical settings for preventing ischemic stroke. However, due to the contraindications (cardiopathy), serious side effects (hemorrhage or cardiac failure) and narrow therapeutic time window, only approximately 5% of patients full benefit from those drugs. More importantly, despite the fact that stroke is related to complex mechanisms in the brain, most pre-clinical trials were often performed using a single agent against single purported mechanism of action specifically targeting the stroke, which led to these efforts failing to provide significant benefit in clinical trials [6]. More effective and safe therapeutic strategies for the prevention and treatment of ischemic stroke are urgently needed. To broaden treatment targets and raise the cure rate, current opinion supports to develop an agent with multiple therapeutic targets.

*Dioscorea Panthaica* Prain et Burkill, as a clinical drug with a long history in China, was extracted from the rhizome of *Dioscorea panthaica* Prain et Burk and *Dioscorea nipponica Makino*, promoting blood circulation, removing blood stasis as well as anti-thrombosis. Diosgenin, as one of the important steroidal saponins in *Dioscorea Panthaica* Prain et Burkill, has a primary role in the synthesis of steroid hormones, contraceptives and cortisone, et al. [7,8,9,10]. Diosgenin has been reported a wide range of pharmacological activities, such as anti-inflammatory, anticancer, antiapoptotic, antihyperglycemia, antioxidant and antihypercholesterolemia activities [11,12,13,14,15]. Furthermore, modern pharmacological investigations have shown that diosgenin not only has significant effects of anti-thrombotic, anticoagulation but also protects neurons damaged by cerebral ischemia reperfusion [16,17,18,19]. Therefore, it has attracted increasing attention because of its great potential to reduce the risk of ischemic stroke by anti-thrombosis and neuroprotective activities. However, the low oral bioavailability due to the lipid solubility of diosgenin has become a major factor limiting its application [20,21]. It is noticeable that several reports have revealed that introducing some units at the C3 position with ester or ether bond significantly improved solubility and diverse physicochemical properties of diosgenin while maintaining the original advantages [22,23,24,25]. Amino acids, as an important organic compounds self-possessed of amine (-NH_2_) and carboxylic (-COOH) functional groups along with side chain particular to each amino acid [26], possess several advantages involving enhancing the selectivity, permeability as well as solubility and reducing the toxicity when it was introduced into a insoluble compounds [27,28,29,30,31]. Besides, amino acids successfully gained our attention because of its neuroprotection. For instance, L-type lysine could improve the function of central nervous tissue [32], glycine can reduce the damage of toxic substances generated during the ischemic processes and generate neuroprotective effects [33] and the neuroprotective effect of sarcosine is associated with changes in glycine transport and reduction of NR2B-containing NMDAR expression [34,35]. Therefore, they have inspired our interest in using diosgenin as the template parent to synthesize novel neuroprotective agents by combination with amino acids. 

From the above, our present experiment was conducted to introduce different kinds of amino acids and obtained compounds with dual action on neuroprotection and angiogenesis. We successfully synthesized eighteen diosgenin-amino acid derivatives, characterizing the synthetic diosgenin-amino acid conjugation by ^1^H-NMR, ^13^C-NMR and HRMS, respectively. Moreover, the neuroprotective effect on SH-SY5Y (human neuroblastoma) cell line and biosafety of H9c2 (human cardiac myocytes) cell line were evaluated in vitro. In addition, Giemsa staining and DAPI staining were performed to observe the neuroprotective mechanism of the compounds with the best activity (DG-15). DG-15 also showed a good angiogenesis effect in the chorioallantoic membrane model (CAM). The results indicated that the major portion of derivatives showed a superior neuroprotective effect than raw materials and DG-15 has double efficacy of neuroprotection and angiogenesis.

## 2. Results

### 2.1. Chemical Synthesis

As shown in Scheme 1, all the designed derivatives were carried out in the following way. Eighteen diosgenin-amino acids derivatives were synthesized by esterification reaction and deprotection reaction, eight of which were novel and had not been previously reported.

We first chose diosgenin as a basic core and introduced nine different kinds of amino acids with protecting groups (Boc group or Cbz group) into its C-3 hydroxyl through the esterification reaction to obtain compounds **1**–**9** and the yield was 50–90%. Subsequently, compounds **1**–**9** were further treated in dry dichloromethane (DCM) with trifluoroacetic (TFA), then the protecting groups were taken off and the compounds **10**–**18** were produced with 50–70%. Table 1 shows the structure of all derivatives. The structures of all target compounds were determined by ^1^H-NMR, ^13^C-NMR and melting points, as well as by a detail HRMS analysis. Also, analysis by the optical rotation test demonstrated that the configuration of the derivatives **1**–**18** was consistent with the results of NMR.

### 2.2. Biological Activities

#### 2.2.1. Neuroprotective Activity Test Using SH-HY5Y Cell

A growing body of evidence supported that the occurrence of diverse brain diseases is closely related to oxidative stress [36] and the SH-SY5Y cells are recognized as a common model for evaluating the function of nerve cells [37]. In order to explore the neuroprotective activity of diosgenin-amino acids derivatives [38,39,40,41,42], the model of 2,4,5-Trihydroxybutyrophenone (TBHP)-induced neurotoxicity in SH-SY5Y cells has been used as an in vitro research platform in this study. The results were expressed as EC_50_ values in Table 2. As shown below, compared with the raw material diosgenin (EC_50_ > 40 μM), the neuroprotective effect of most compounds was distinctly enhanced. From an overall perspective, introduction of amino acid (without Boc group) could display lower EC_50_ values than the rest of compounds that were protected by the Boc group. For example, the EC_50_ of DG-13 and DG-15 in separate improved over 3 times and 2 times than DG-4 and DG-6. Among them, compounds DG-3, 5, 13, 15 and 16 (EC_50_ = 15.20 ± 0.96; 9.51 ± 1.79; 7.22 ± 1.58; 6.86 ± 0.69; 11.38 ± 2.62 μM, respectively) performed well and acted more in neuroprotection than the positive drug edaravone (EC_50_ = 21.60 ± 3.04 μM). In order to better select high-efficiency as well as low-toxic compounds from all the compounds, it is necessary to conduct biosafety evaluation. Therefore, compounds DG-13 and 15 with optimal activity were selected for further cytotoxicity evaluation.

#### 2.2.2. Biosafety Evaluation Using H9c2 Cell

The cytotoxicity of those superior derivatives (DG-13 and 15) in vitro were verified on rat embryonic cardiomyocytes (H9c2) by the MTT assay. The cell viability was summarized in Figure 1. It was found diosgenin was toxic under all treated groups. For example, cell viability was nearly 60% even under the concentration of 50 μM. While the cytotoxicity of the modified derivatives was significantly different. It was observed that the cytotoxicity of DG-13 was significantly enhanced compared with diosgenin and the survival rate was only 12% when the concentration was 25 μM. On the contrary, DG-15 had no inhibitory effect on cells at concentrations below 25 μM, which was better than that of diosgenin. Compound DG-13 and 15 have similar neuroprotective activities on SH-SY5Y, however their cytotoxicity is quite different. Thus, compound DG-15 with high effects and low poison was selected as a target compound for more in-depth research.

#### 2.2.3. Morphological Observation Using Giemsa and DAPI Staining

The morphological change in SH-SY5Y cells induced by TBHP was analyzed by Giemsa and DAPI Staining. As we can see from Figure 2, compared with the control group (no treatment addition to the wells), SH-SY5Y cells treatment with TBHP showed an obvious changes in morphology and quantity, such as shrinkage of the cell bodies, appearance of nuclear fragmentation and disappearance of reticular formation, which suggested that the model of cell is valid. After treating damaged SH-SY5Y cells with 10, 20, 40 μM of DG-15, the typical signs for apoptosis were reduced. With increasing concentration of DG-15, the number of cells increased significantly and the promotion of cell synapse formation obviously. The result indicated that DG-15 has a protective effect on SH-SY5Y cells damaged by TBHP in proper concentration range. Similarly, as shown in Figure 3, in three groups (10, 20, 40 μM) the number of SH-SY5Y cells was increasingly larger and the cells of uniform size with apoptosis gradually reduced than the control group. Based on the above results, we found that the compound DG-15 could lead to an alleviated morphological lesion to injured SH-SY5Y cells at lower concentrations in vitro.

#### 2.2.4. Promotes Angiogenesis in CAM Assay

The CAM model was used to further investigate the effects of DG-15 in angiogenesis [43,44,45,46,47]. The number of new vessels (inner diameter < 100 μm) radiating from the gelatin sponge were counted [48,49,50]. Macroscopic observation shown that the new blood vessels grow radially around the gelatin sponge after administration (Figure 4) and DG-15 could dramatically promote small angiogenesis in a dose dependent manner on CAM. As shown in Figure 5, the number of blood vessels increased by 60.61% and 80.30% at low and high concentrations (1 mg/mL and 4 mg/mL) of DG-15, respectively. The result implied that compound DG-15 has a certain role in promoting angiogenesis at high concentrations.

## 3. Discussion

The treatment of ischemic stroke is still a worldwide problem. Research has focused on the repair of neurons. Also, recent pharmacological studies illustrated angiogenesis might be a crucial determinant of repairing damaged neurons after stroke and demonstrated a correlation with survival rate of stroke patients [41,42,43,44,45,46,47,48,49,50,51,52,53,54,55]. Inspired by biological characteristics of them, we introduced nine amino acids into diosgenin to enhance the function of neuroprotection and angiogenesis in this study. While some amino acids play essential roles in neuroprotection, it is a pity that the yield of these analogues was low than we expected. For instance, histidine (His) possess a relevant role in neuroprotection by reacting with copper ions and zinc, having an impact on proteasome functions and polyubiquitination reactions [56,57]. The effect of His was also ascribed to the inhibition of excessive L-glutamine transport to mitochondria [58], which was the reason of connection between the neuroprotective effects of His and strokes caused by oxidative stress. In addition, the results of neuroprotective activity evaluation indicated that the introduction of lysine, an alkaline amino acid, produced the higher cytotoxicity of compound (DG-18) than other compounds with neutral amino acids. Similar cytotoxicity occurs in many antitumor drugs with lysine residues [59]. Preliminary structure-activity relationships analysis indicated that, the derivatives containing Cbz group represented better neuroprotective activity than the compounds that have been stripped of its protective group (EC_50_: DG-8 < DG-17; DG-9 < DG-18). This result was opposite of those of the BOC-protected compound.

## 4. Materials and Methods

### 4.1. Materials and Instruments

Diosgenin was purchased from KONO CHEM Co., Ltd. (Xian, China). Amino acid (Boc-Gly, Boc-L-Ala, Boc-Sar, Boc-L-Pro, Boc-L-Leu, Boc-L-Ile, Boc-L-Phe, Cbz-L-Val and Cbz-L-Lys) were obtained from Beijing Enochai Technology Co., Ltd. (Beijing, China), all reagents were required from suppliers with no purification measures. ^1^H-NMR and ^13^C-NMR spectra were recorded in CDCl_3_ or Acetone-*d_6_* solutions on a Bruker-500 spectrometer (Bruker, Dresden, Germany) with tetramethylsilane (TMS; TCI, Tokyo, Japan) as an internal reference. Mass spectra were acquired using a Thermo Scientific TMLTQ Orbitrap XL hybrid FTMS instrument (Thermo Technologies, New York, NY, USA) and an X-5 micro melting point apparatus (Beijing Tech Instrument Co., Ltd., Beijing, China). Giemsa, 6-diamidino-2-phenylindole (DAPI) were provided by Beijing Solarbio Science & Technology Co. Ltd. (Beijing, China). SH-SY5Y cells and H9c2 cells were purchased from Institute of Peking Union Medical College (Beijing, China). Cellular morphologies were observed using an inverted fluorescence microscope (Olympus CKX53, Tokyo, Japan).

### 4.2. Chemical Syntheses

#### 4.2.1. The procedure for esterification at C3-OH (method 1)

The compound diosgenin (1 equiv.) and the protected amino acid with Boc or Cbz group (1.2 equiv.) were dissolved in dry dichloromethane (DCM, 10 mL), the 1-Ethyl-3-(3-dimethylaminopropyl) carbodiimide hydrochloride (EDCI, 1.5 equiv.) and the 4-dimethylaminopyridine (DMAP, 0.5 equiv.) were added subsequently. Then the mixture was stirred at room temperature for 12 h. Next, the solution was extracted 3 times with DCM to obtain crude product. Before purifying the crude produced by flash chromatography (silica gel, petroleum ether: acetone 10:1), the organic layer was dried by anhydrous sodium sulfate and evaporated under vacuum. The target compounds DG-1 to 9 were obtained.

#### 4.2.2. The procedure for deprotection reaction (method 2)

After solving the diosgenin-amino acids derivatives DG-1 to 9 (0.33 equiv.) in dry DCM (1 equiv.) respectively, the trifluoroacetic acid (TFA, 0.1 equiv.) was added. The reaction was carried out in ice-water at 4 °C for 4 h and continuously stirred. Then the reaction mixture was evaporated under vacuum and diluted with DCM, then successively washed with saturated sodium bicarbonate solution and water. Finally, the organic layer was collected, dried and separated by flash chromatography with dichloromethane-acetone (10:1). The ^1^H-NMR and ^13^C-NMR spectra of compound **1**–**18** can be seen in the Appendix A.

3β-(*N*-Boc-glycine)-diosgenin (*Compound DG-1*). Compound DG-1 was obtained as white powder by method 1, yield: 78.5%; m.p.: 218.7 °C ${\left[\alpha \right]}_{D}^{25}$ = −89.72° (c = 1.0 mg/mL, CHCl_3_); ^1^H-NMR (500 MHz, Acetone-*d*_6_): *δ* (ppm) 6.22 (t, *J* = 5.5 Hz, 1H), 5.40 (brs, 1H), 4.60–4.54 (m, 1H), 4.40–4.35 (m, 1H), 3.77 (d, *J* = 6.0 Hz, 2H), 3.43–3.40 (m, 1H), 3.29 (t, *J* = 10.5 Hz, 1H), 2.32 (d, *J* = 8.0 Hz, 2H), 1.42 (brs, 9H, Boc-H), 1.07(s, 3H), 0.96 (d, *J* = 7.0 Hz, 3H), 0.82 (s, 3H), 0.77 (d, *J* = 6.5 Hz, 3H), 1.00–2.00 (m, 22H, Sterane Structure); ^13^C-NMR (125 MHz, Acetone-*d*6): *δ* (ppm) 170.62 (C1′), 156.96 (C1″), 140.81 (C5), 123.25 (C6), 109.69 (C22), 81.59 (C16), 79.38 (C2″), 75.15 (C3), 67.38 (C26), 63.67, 57.40, 51.15, 43.33, 42.55, 40.60, 39.00, 37.91, 32.92, 32.72, 32.40, 31.28, 30.31, 30.16, 30.01, 29.85, 29.70, 28.73, 21.73, 19.82, 17.64, 16.83, 15.21; HRMS (ESI) *m*/*z*: 572.3941 [M + H]^+^, calcd for: C_34_H_53_O_6_N 571.3941.

3β-(*N*-Boc-l-alanine)-diosgenin (*Compound DG-2*). Compound DG-2 was obtained as white powder by method 1, yield: 78.6%; m.p.: 218.9 °C ${\left[\alpha \right]}_{D}^{25}$ = −86.96° (c = 1.0 mg/mL, CHCl_3_); ^1^H-NMR (500 MHz, CDCl_3_): *δ* (ppm) 5.37 (brs, 1H), 4.65 (s, 1H), 4.38–4.42 (m, 1H), 3.93 (s, 1H), 3.85 (s, 1H), 3.46 (d, *J* = 10.5 Hz, 1H), 3.36 (t, *J* = 11.0 Hz, 1H), 2.91 (d, *J* = 11.5 Hz, 3H), 2.33 (s, 2H), 2.16 (m, 2H), 1.44 (brs, 9H, Boc-H), 1.02 (s, 3H), 0.97 (d, *J* = 7.0 Hz, 3H), 0.78 (s, 6H), 1.00–2.00 (m, 20H, Sterane Structure); ^13^C-NMR (125 MHz, CDCl_3_): *δ* (ppm) 169.46 (C1′), 155.64 (C1″), 139.53 (C5), 122.78 (C6), 109.40 (C22), 80.93 (C16), 80.15 (C2″), 74.80 (C3), 66.97 (C26), 62.21, 56.55, 51.45, 50.65, 50.04, 41.74, 40.38, 39.84, 38.23, 37.02, 36.84, 35.71, 32.16, 31.97, 31.52, 31.06, 30.43, 28.94, 28.43, 27.87, 20.94, 19.45, 17.27, 16.41, 14.65. HRMS (ESI) *m*/*z*: 586.9252 [M + H]^+^, calcd for: C_35_H_55_O_6_N 585.9252.

3β-(*N*-Boc-sarcosine)-diosgenin (*Compound DG-3*). Compound DG-3 was obtained as white powder by method 1, yield: 81.2%; m.p.: 204.0 °C, ${\left[\alpha \right]}_{D}^{25}$ = −99.20° (c = 1.0 mg/mL, CHCl_3_); ^1^H-NMR (500 MHz, CDCl_3_): *δ* (ppm) 5.37 (brs, 1H), 4.67–4.63 (m, 1H), 4.38–4.42 (m, 1H), 3.46 (d, *J* = 8.5 Hz, 1H), 3.36 (t, *J* = 11.0 Hz, 1H), 2.31 (d, *J* = 8.0 Hz, 2H), 2.16 (m, 2H), 1.44 (brs, 9H, Boc-H), 1.36 (d, *J* = 7.0 Hz, 3H), 1.03 (s, 3H), 0.96 (d, *J* = 6.5 Hz, 3H), 0.78 (s, 6H), 1.00–2.00 (m, 22H, Sterane Structure); ^13^C-NMR (125 MHz, CDCl_3_): *δ* (ppm) 172.92 (C1′), 155.23 (C1″), 139.57 (C5), 122.71 (C6), 109.41 (C22), 80.92 (C16), 74.98 (C3), 66.97 (C26), 62.21, 56.56, 50.05, 49.49, 41.74, 40.39, 39.85, 38.03, 37.01, 36.85, 32.17, 31.97, 31.52, 30.43, 28.94, 28.47, 27.77, 20.94, 19.46, 18.94, 17.27, 16.41, 14.65. HRMS (ESI) *m*/*z*: 586.4225 [M + H]^+^, calcd for: C_35_H_55_O_6_N 585.4225.

3β-(*N*-Boc-l-proline)-diosgenin (*Compound DG-4*). Compound DG-4 was obtained as white powder by method 1, yield: 80.2%; m.p.: 183.4 °C, ${\left[\alpha \right]}_{D}^{25}$ = −106.67° (c = 1.0 mg/mL, CHCl_3_); ^1^H-NMR (500 MHz, CDCl_3_): *δ* (ppm) 5.37 (brs, 1H), 4.62 (d, *J* = 9.0 Hz, 1H), 4.38–4.43 (m, 1H), 4.17–4.28 (m, 1H), 2.31 (d, *J* = 7.0 Hz, 2H), 2.16 (m, 2H), 1.44 (brs, 9H, Boc-H), 1.03 (d, *J* = 5.0Hz, 3H), 0.97 (d, *J* = 7.0 Hz, 3H), 0.78 (s, 6H), 1.00–2.00 (m, 28H, Sterane Structure); ^13^C-NMR (125 MHz, CDCl_3_): *δ* (ppm) 172.73 (C1′), 154.03 (C1″), 139.61 (C5), 122.70 (C6), 109.41 (C22), 80.93 (C16), 74.48 (C3), 66.98 (C26), 62.22, 59.40, 56.55, 50.06, 46.70, 46.48, 41.75, 40.39, 39.85, 37.06, 32.18, 31.98, 31.53, 31.12, 30.44, 30.12, 28.94, 28.59, 28.52, 27.77, 24.40, 23.68, 20.95, 19.48, 17.27, 16.41, 14.66. HRMS (ESI) *m*/*z*: 612.4243 [M + H]^+^, calcd for: C_37_H_57_O_6_N 611.4243.

3β-(*N*-Boc-l-leucine)-diosgenin (*Compound DG-5*). Compound DG-5 was obtained as white powder by method 1, yield: 86.3%; m.p.: 150.6 °C ${\left[\alpha \right]}_{D}^{25}$ = −76.77° (c = 1.0 mg/mL, CHCl_3_); ^1^H-NMR (500 MHz, CDCl_3_): *δ* (ppm) 5.37 (brs, 1H), 4.88 (d, *J* = 8.5 Hz, 1H), 4.62–4.69 (m, 1H), 4.40–4.45 (m, 1H), 4.26–4.28 (m, 1H), 3.47–3.50 (m, 1H), 3.39 (t, *J* = 11.0 Hz, 1H), 2.33 (d, *J* = 7.5 Hz, 2H), 2.01 (m, 2H), 1.46 (brs, 9H, Boc-H), 1.06 (s, 3H), 0.95–1.00 (m, 9H), 0.80 (t, *J* = 3 Hz, 6H), 1.00–2.00 (m, 23H, Sterane Structure); ^13^C-NMR (125 MHz, CDCl_3_): *δ* (ppm) 173.04 (C1′), 155.55 (C1″), 139.65 (C5), 122.66 (C6), 109.41 (C22), 80.94 (C16), 79.81 (C2″), 74.87 (C3), 66.98 (C26), 62.22, 56.57, 52.38, 50.06, 42.10, 41.75, 40.39, 39.86, 38.09, 37.03, 36.86, 32.18, 31.98, 31.53, 30.44, 28.95, 28.47, 27.81, 24.95, 22.99, 22.15, 20.95, 19.48, 17.28, 16.42, 14.66. HRMS (ESI) *m*/*z*: 628.4568 [M + H]^+^, calcd for: C_38_H_61_O_6_N 627.4568.

3β-(*N*-Boc-l-isoleucine)-diosgenin (*Compound DG-6*). Compound DG-6 was obtained as white powder by method 1, yield: 82.4%; m.p.: 161.4 °C, ${\left[\alpha \right]}_{D}^{25}$ = −75.79° (c = 1.0 mg/mL, CHCl_3_); ^1^H-NMR (500 MHz, CDCl_3_): *δ* (ppm) 5.37 (brs, 1H), 5.03 (d, *J* = 8.5 Hz, 1H), 4.63–4.67 (m, 1H), 4.38–4.43 (m, 1H), 3.45–3.48 (m, 1H), 3.37 (t, *J* = 11.0 Hz, 1H), 2.30–2.35 (m, 2H), 1.95–2.01 (m, 2H), 1.44 (brs, 9H, Boc-H), 1.04 (s, 3H), 0.97 (d, *J* = 7.0 Hz, 3H), 0.90–0.94 (m, 6H), 0.78 (t, *J* = 3.0 Hz), 1.00–2.00 (m, 24H, Sterane Structure); ^13^C-NMR (125 MHz, CDCl_3_): *δ* (ppm) 171.82 (C1′), 155.71 (C1″), 139.60 (C5), 122.74 (C6), 109.42 (C22), 80.94 (C16), 79.76 (C2″), 74.90 (C3), 66.99 (C26), 62.22, 58.04, 56.57, 50.07, 41.75, 40.40, 39.86, 38.33, 38.22, 37.03, 36.86, 32.18, 31.98, 31.52, 30.44, 28.95, 28.49, 27.88, 25.25, 20.95, 19.48, 17.28, 16.42, 15.64, 14.66, 11.90. HRMS (ESI) *m*/*z*: 628.4563 [M + H]^+^, calcd for: C_38_H_61_O_6_N 627.4563.

3β-(*N*-Boc-l-phenylalanine)-diosgenin (*Compound DG-7*). Compound DG-7 was obtained as white powder by method 1, yield: 80.1%; m.p.: 163.4 °C, ${\left[\alpha \right]}_{D}^{25}$ = −84.03° (c = 1.0 mg/mL, CHCl_3_); ^1^H-NMR (500 MHz, CDCl_3_): *δ* (ppm) 7.14–7.30 (m, 5H), 5.36 (brs, 1H), 4.97 (d, *J* = 8.5 Hz, 1H), 4.59–4.63 (m, 1H), 4.54–4.58 (m, 1H), 4.39–4.53 (m, 1H), 3.46–3.48 (m, 1H), 3.37 (t, *J* = 11 Hz, 1H), 3.03–3.11 (m, 2H), 2.22–2.25 (m, 2H), 1.96–2.02 (m, 2H), 1.42 (brs, 9H, Boc-H), 1.02 (s, 3H), 0.97 (d, *J* = 7.0 Hz, 6H), 0.79 (d, *J* = 4.5 Hz, 6H), 1.00–2.00 (m, 20H, Sterane Structure); ^13^C-NMR (125 MHz, CDCl_3_): *δ* (ppm) 171.41(C1′), 155.21 (C1″), 139.60 (C5), 136.27 (Ar-C), 129.59 (Ar-C), 128.59 (Ar-C), 127.08 (Ar-C), 122.74(C6), 109.41 (C22), 80.93 (C16), 79.92 (C2″), 75.21 (C3), 66.98 (C26), 62.22, 56.55, 54.66, 50.05, 41.75, 40.39, 39.84, 38.59, 38.01, 37.00, 36.85, 32.18, 31.98, 31.53, 30.44, 28.95, 28.45, 27.78, 20.94, 19.45, 17.28, 16.41, 14.66. HRMS (ESI) *m*/*z*: 662.4389 [M + H]^+^, calcd for: C_41_H_59_O_6_N 661.4389.

3β-(*N*-Cbz-l-valine)-diosgenin (*Compound DG-8*). Compound DG-8 was obtained as white powder by method 1, yield: 68.6%; m.p.: 158.4 °C, ${\left[\alpha \right]}_{D}^{25}$ = −80.00° (c = 1.0 mg/mL, CHCl_3_); ^1^H-NMR (500 MHz, CDCl_3_): *δ* (ppm) 7.31–7.37 (m, 5H), 5.37 (brs, 1H), 5.27 (d, *J* = 9 Hz, 1H), 5.11 (s, 2H), 4.67 (d, *J* = 9.5 Hz, 1H), 4.39–4.43 (m, 1H), 4.25–4.28 (m, 1H), 3.46–3.49 (m, 1H), 3.37 (t, *J* = 11.0 Hz, 1H), 2.31 (d, *J* = 7.0 Hz, 2H), 1.96–2.03 (m, 2H), 1.04 (s, 3H), 0.97 (d, *J*
*=* 7.0 Hz, 6H), 1.00–2.00 (m, 21H, Sterane Structure); ^13^C-NMR (125 MHz, CDCl_3_): *δ* (ppm) 171.50 (C1′), 156.37 (C1″), 139.50 (C5), 128.68 (Ar-C), 128.32 (Ar-C), 122.81 (C6), 109.42 (C22), 80.94 (C16), 75.11 (C3), 67.11 (C2″), 66.99 (C26), 62.22, 59.16, 56.57, 50.06, 41.76, 40.40, 39.85, 38.18, 37.00, 36.85, 32.18, 31.98, 31.56, 31.52, 30.44, 28.95, 27.86, 20.95, 19.47, 19.08, 17.62, 17.28, 16.42, 14.66. HRMS (ESI) *m*/*z*: 648.4270 [M + H]^+^, calcd for: C_40_H_57_O_6_N 647.4270.

3β-(*N*-Cbz-l-lysine)-diosgenin (*Compound DG-9*). Compound DG-9 was obtained as white powder by method 1, yield: 68.6%; m.p.: 135.8 °C, ${\left[\alpha \right]}_{D}^{25}$ = −43.56° (c = 1.0 mg/mL, CHCl_3_); ^1^H-NMR (500 MHz, CDCl_3_): *δ* (ppm) 7.30–7.35 (m, 10H), 5.36 (brs, 1H), 5.08–5.12 (m, 4H), 4.65 (d, *J* =10.0 Hz, 1H), 4.39–4.43 (m, 1H), 4.29–4.32 (m, 1H), 3.46–3.48 (m, 1H), 3.38 (t, *J* = 11.0 Hz, 1H), 3.19 (d, *J* = 5.5 Hz, 2H), 2.29–2.30 (m, 2H), 1.96–2.00 (m, 2H), 1.43 (s, 6H), 1.03 (s, 3H), 0.97–0.98 (m, 3H), 0.79 (d, *J* = 4.0 Hz, 6H), 1.00–2.00 (m, 22H, Sterane Structure); ^13^C-NMR (125 MHz, CDCl_3_): *δ* (ppm) 171.88 (C1′), 156.60 (C1″), 156.09 (C1″′), 139.41 (C5), 136.71 (Ar-C), 136.40 (Ar-C), 128.65 (Ar-C), 128.22 (Ar-C), 122.87 (C6), 109.42 (C22), 80.93 (C16), 75.29 (C3), 67.12 (C2″), 66.99 (C2″′), 66.79 (C26), 62.21, 56.56, 53.87, 50.05, 41.75, 40.73, 40.39, 39.85, 38.07, 36.99, 36.83, 32.52, 32.17, 31.98, 31.51, 30.44, 29.50, 28.95, 27.80, 27.05, 22.33, 20.95, 19.46, 17.28, 16.42, 14.66. HRMS (ESI) *m*/*z*: 811.4846 [M + H]^+^, calcd for: C_49_H_66_O_8_N_2_ 810.4846.

3β-(glycine)-diosgenin (*Compound DG-10*). Compound DG-10 was obtained as white powder by method 2, yield: 56.7%; m.p.: 110.8 °C, ${\left[\alpha \right]}_{D}^{25}$ = −82.05° (c = 1.0 mg/mL, CHCl_3_); ^1^H-NMR (500 MHz, Acetone-*d*6): *δ* (ppm) 5.40 (brs, 1H), 4.54–4.60 (m, 1H), 4.35–4.40 (m, 1H), 3.40–3.43 (m, 1H), 3.30 (t, *J* = 11.0 Hz, 1H), 2.34 (d, *J* = 6.0 Hz, 2H), 2.09 (s, 2H), 1.08 (s, 3H), 0.96 (d, *J* = 7.0 Hz, 3H), 0.83 (s, 3H), 0.76 (d, *J* = 6.5 Hz), 1.00–2.00 (m, 24H, Sterane Structure); ^13^C-NMR (125 MHz, Acetone-*d*6): *δ* (ppm) 170.46 (C1′), 140.77 (C5), 123.01 (C6), 109.53 (C22), 81.44 (C16), 74.53 (C3), 67.22 (C26), 63.50, 57.24, 51.01, 42.39, 41.01, 40.44, 38.89, 37.81, 37.52, 32.76, 32.55, 32.30, 32.24, 31.12, 28.51, 21.57, 19.67, 17.47, 16.65, 15.04. HRMS (ESI) *m*/*z*: 472.3399 [M + H]^+^, calcd. for: C_29_H_45_O_4_N 471.3399.

3β-(l-alanine)-diosgenin. (*Compound DG-11*). Compound DG-11was obtained as white powder by method 2, yield: 60.1%; m.p.: 219.1 °C, ${\left[\alpha \right]}_{D}^{25}$ = −77.79° (c = 1.0 mg/mL, CHCl_3_); ^1^H-NMR (500 MHz, CDCl_3_): *δ* (ppm) 5.37 (brs, 1H), 4.60–4.65 (m, 1H), 4.38–4.43 (m, 1H), 3.48–3.53 (m, 1H), 3.45–3.47 (m, 1H), 3.37 (t, *J* = 11.0 Hz, 1H), 2.31 (d, *J* = 7.5 Hz, 2H), 1.95–2.02 (m, 2H), 1.85–1.87 (m, 2H), 1.33 (d, *J* = 7.0 Hz, 3H), 1.04 (s, 3H), 0.97 (d, *J* = 7.0 Hz, 3H), 0.78 (t, *J* = 3.0 Hz, 6H), 1.00–2.00 (m, 20H, Sterane Structure); ^13^C-NMR (125 MHz, CDCl_3_): *δ* (ppm) 176.06(C1′), 139.66 (C5), 122.65 (C6), 109.41 (C22), 80.93 (C16), 74.51 (C3), 66.98 (C26), 62.22, 56.57, 50.30, 50.07, 41.75, 40.39, 39.86, 38.15, 37.06, 36.86, 32.18, 31.98, 31.53, 30.44, 28.94, 27.83, 20.96, 20.80, 19.47, 17.27, 16.42, 14.66. HRMS (ESI) *m*/*z*: 486.3554 [M + H]^+^, calcd for: C_30_H_47_O_4_N 485.3554.

3β-(sarcosine)-diosgenin (*Compound DG-12*). Compound DG-12 was obtained as white powder by method 2, yield: 68.5%; m.p.: 176.0 °C, ${\left[\alpha \right]}_{D}^{25}$ = −78.57° (c = 1.0 mg/mL, CHCl_3_); ^1^H-NMR (500 MHz, CDCl_3_): *δ* (ppm) 5.37 (brs, 1H), 4.64–4.70 (m, 1H), 4.38–4.42 (m, 1H), 3.44–3.47 (m, 1H), 3.34–3.38 (m, 3H), 2.44 (s, 3H), 2.33 (d, *J* = 7.5 Hz, 2H), 2.00 (s, 1H), 1.84–1.87 (m, 2H), 1.03 (s, 3H), 0.96 (d, *J* = 7.0 Hz, 3H), 0.78 (t, *J* = 6.0 Hz, 6H), 1.00–2.00 (m, 20H, Sterane Structure); ^13^C-NMR (125 MHz, CDCl_3_): *δ* (ppm) 171.71(C1′), 139.64 (C5), 122.67 (C6), 109.39 (C22), 80.91 (C16), 74.53 (C3), 66.96 (C26), 62.21, 56.55, 52.94, 50.05, 41.73, 40.38, 39.84, 38.22, 37.04, 36.84, 36.15, 32.16, 31.96, 31.52, 30.42, 28.93, 27.88, 20.94, 19.45, 17.26, 16.40, 14.65. HRMS (ESI) *m*/*z*: 486.3545 [M + H]^+^, calcd for: C_30_H_47_O_4_N 485.3545.

3β-(l-proline)-diosgenin (*Compound DG-13*). Compound DG-13 was obtained as white powder by method 2, yield: 62.9%; m.p.: 195.2 °C, ${\left[\alpha \right]}_{D}^{25}$ = −83.52° (c = 1.0 mg/mL, CHCl_3_); ^1^H-NMR (500 MHz, CDCl_3_): *δ* (ppm) 5.37 (brs, 1H), 4.61–4.66 (m, 1H), 4.38–4.43 (m, 1H), 3.71–3.74 (m, 1H), 3.45–3.48 (m, 1H), 3.37 (t, *J* = 11.0 Hz, 1H), 3.08–3.10 (m, 1H), 2.89–2.91 (m, 1H), 2.32(d, *J* = 7.5 Hz, 2H), 2.11–2.15 (m, 1H), 1.95–2.01 (m, 2H), 1.03(s, 3H), 0.97 (d, *J* = 7.0 Hz, 3H), 0.78(t, *J* = 3.0 Hz, 6H), 1.00–2.00 (m, 24H, Sterane Structure); ^13^C-NMR (125 MHz, CDCl_3_): δ (ppm) 174.98 (C1′), 139.68 (C5), 122.63 (C6), 109.41 (C22), 80.93 (C16), 74.58 (C3), 66.98 (C26), 62.22, 60.04, 56.56, 50.06, 47.22, 41.75, 40.39, 39.85, 38.15, 37.06, 36.86, 32.18, 31.98, 31.53, 30.56, 30.43, 28.94, 27.83, 25.62, 20.95, 19.48, 17.27, 16.41, 14.66. HRMS (ESI) *m*/*z*: 512.3701 [M + H]^+^, calcd for: C_32_H_49_O_4_N 511.3701.

3β-(l-leucine)-diosgenin (*Compound DG-14*). Compound DG-14 was obtained as white powder by method 2, yield: 66.5%; m.p.: 154.4 °C, ${\left[\alpha \right]}_{D}^{25}$ = −94.55° (c = 1.0 mg/mL, CHCl_3_); ^1^H-NMR (500 MHz, Acetone-*d*6): *δ* (ppm) 5.40 (brs, 1H), 4.51–4.55 (m, 1H), 4.35–4.40 (m, 1H), 4.09–4.11 (m, 1H), 3.40–3.43 (m, 1H), 3.30 (t, *J* = 10.75 Hz, 1H), 2.32 (d, *J* = 7.5 Hz, 2H), 2.09 (s, 1H), 1.07 (s, 3H), 0.96 (d, *J* = 7.0 Hz, 3H), 0.92 (d, *J* = 6.0Hz, 3H), 0.86 (d, *J* = 7.0 Hz, 3H), 0.82 (s, 3H), 0.76 (d, *J* = 6.5 Hz), 1.00–2.00 (m, 26H, Sterane Structure); ^13^C-NMR (125 MHz, Acetone-*d*6): *δ* (ppm) 172.43 (C1′), 140.88 (C5), 123.17 (C6), 109.67 (C22), 81.58 (C16), 74.60 (C3), 67.37 (C26), 63.64, 62.89, 57.38, 51.14, 43.36, 42.53, 41.15, 40.58, 39.00, 37.93, 37.66, 32.90, 32.69, 32.38, 31.26, 28.62, 25.63, 23.48, 22.57, 21.71, 19.83, 17.62, 16.80, 15.19. HRMS (ESI) *m*/*z*: 528.4016 [M + H]^+^, calcd for: C_33_H_53_O_4_N 527.4016.

3β-(l-isoleucine)-diosgenin (*Compound DG-15*). Compound DG-15 was obtained as white powder by method 2, yield: 70.2%; m.p.: 147.1 °C, ${\left[\alpha \right]}_{D}^{25}$ = −76.34° (c = 1.0 mg/mL, CHCl_3_); ^1^H-NMR (500 MHz, Acetone-d6): δ (ppm) 5.41 (brs, 1H), 4.56–4.58 (m, 1H), 4.37–4.40 (m, 1H), 3.81–3.83 (m, 1H), 3.41–3.43 (m, 1H), 3.30 (t, *J* = 11.0 Hz), 2.31 (d, *J* = 6.5 Hz, 2H), 2.09 (s, 1H), 1.07 (s, 3H), 0.96 (d, *J* = 7.0 Hz, 3H), 0.85–0.97 (m, 6H), 0.83 (s, 3H), 0.77 (d, *J* = 6.5 Hz, 3H), 1.00–2.00 (m, 26H, Sterane Structure); ^13^C-NMR (125 MHz, Acetone-*d*6): *δ* (ppm) 171.65 (C1′), 140.84 (C5), 123.17 (C6), 109.65 (C22), 81.56 (C16), 74.46 (C3), 69.76, 67.35 (C26), 63.62, 60.38, 57.36, 51.12, 42.52, 41.13, 40.56, 39.25, 39.06, 37.91, 37.65, 32.89, 32.68, 32.37, 31.25, 28.65, 25.91, 21.70, 19.81, 17.61, 16.79, 16.11, 15.18, 12.01, 11.83. HRMS (ESI) *m*/*z*: 528.4017 [M + H]^+^, calcd for: C_33_H_53_O_4_N.

3β-(l-phenylalanine)-diosgenin (*Compound DG-16*). Compound DG-16 was obtained as white powder by method 2, yield: 71.9%; m.p.: 157.7 °C, ${\left[\alpha \right]}_{D}^{25}$ = −66.67° (c = 1.0 mg/mL, CHCl_3_); ^1^H-NMR (500 MHz, CDCl_3_): *δ* (ppm) δ7.20–7.31 (m, 5H), 5.36 (brs, 1H), 4.60–4.62 (m, 1H), 4.39–4.43 (m, 1H), 3.69 (s, 1H), 3.46–3.48 (m, 1H), 3.37 (t, *J* = 11.0 Hz, 1H), 3.04–3.08 (m, 1H), 2.85–2.89 (m, 1H), 2.23–2.24 (m, 1H), 2.17 (s, 1H), 1.96–2.01 (m, 2H), 1.02 (s, 3H), 0.97 (d, *J* = 7.0 Hz, 3H), 0.78 (brs, 6H), 1.00–2.00 (m, 21H, Sterane Structure); ^13^C-NMR (125 MHz, Acetone-*d*6): *δ* (ppm) 174.56 (C1′), 139.67 (C5), 137.40 (Ar-C), 129.50 (Ar-C), 128.65 (Ar-C), 126.92 (Ar-C), 122.65 (C6), 109.41 (C22), 80.93 (C16), 74.68 (C3), 66.98 (C26), 62.22, 56.55, 56.02, 50.05, 41.74, 41.32, 40.39, 39.84, 38.06, 37.04, 36.85, 32.17, 31.97, 31.53, 30.43, 28.94, 27.83, 20.94, 19.45, 17.27, 16.41, 14.66. HRMS (ESI) *m*/*z*: 562.3854 [M + H]^+^, calcd for: C_36_H_51_O_4_N 561.3854.

3β-(l-valine)-diosgenin (*Compound DG-17*). Compound DG-17 was obtained as white powder by method 2, yield: 58.2%; m.p.: 139.1°C, ${\left[\alpha \right]}_{D}^{25}$ = −69.72° (c = 1.0 mg/mL, CHCl_3_); ^1^H-NMR (500 MHz, CDCl_3_): *δ* (ppm) 5.37 (s, 1H), 4.64–4.66 (m, 1H), 4.38–4.42 (m, 1H), 3.45–3.47 (m, 1H), 3.36 (t, *J* = 11.0 Hz, 1H), 2.32(d, *J* = 6.5 Hz, 2H), 2.14–2.16 (m, 2H), 1.03 (s, 3H), 0.95 (d, *J* = 6.5 Hz, 6H), 0.91 (d, *J* = 7.0 Hz, 3H), 0.78 (s, 6H), 1.00–2.00 (m, 24H, Sterane Structure); ^13^C-NMR (125 MHz, Acetone-*d*6): *δ* (ppm) 174.78 (C1′), 139.64 (C5), 122.67 (C6), 109.41 (C22), 80.93 (C16), 74.44 (C3), 69.41, 66.97 (C26), 62.21, 59.94, 56.56, 50.07, 41.74, 40.38, 39.85, 38.25, 37.06, 36.86, 35.16, 32.17, 31.97, 31.52, 30.43, 28.94, 27.90, 20.94, 19.47, 19.38, 17.34, 17.27, 16.41, 14.65. HRMS (ESI) *m*/*z*: 514.3864 [M + H]^+^, calcd for: C_32_H_51_O_4_N 513.3864.

3β-(l-lysine)-diosgenin (*Compound DG-18*). Compound DG-18 was obtained as white powder by method 2, yield: 70.2%; m.p.: 189.4°C, ${\left[\alpha \right]}_{D}^{25}$ = −63.89° (c = 1.0 mg/mL, CHCl_3_), yield: 64.6%. ^1^H-NMR (500 MHz, CDCl_3_): *δ* (ppm) 5.37 (brs, 1H), 4.61–4.63 (m, 1H), 4.37–4.42 (m, 1H), 3.45–3.47 (m, 1H), 3.34–3.38 (m, 2H), 2.62–2.72 (m, 4H), 2.30 (d, *J* = 7.5 Hz, 2H), 2.16 (s, 2H), 1.03 (s, 3H), 0.96 (d, *J* = 7.0 Hz, 3H), 0.77 (d, *J* = 3.5Hz, 6H), 1.00–2.00 (m, 28H, Sterane Structure); ^13^C-NMR (125 MHz, CDCl_3_): *δ* (ppm) 175.56 (C1′), 139.60 (C5), 122.69 (C6), 109.40 (C22), 80.91 (C16), 74.49 (C3), 66.96 (C26), 62.20, 56.55, 54.50, 50.05, 41.73, 40.37, 39.84, 38.18, 37.04, 36.84, 34.67, 32.16, 31.96, 31.51, 30.41, 28.92, 27.85, 23.01, 20.94, 19.46, 17.26, 16.40, 14.64. HRMS (ESI) *m*/*z*: 543.4121 [M + H]^+^, calcd for: C_33_H_54_O_4_N_2_ 542.4121.

### 4.3. Bio-Evaluation Methods

#### 4.3.1. Cell Culture

SH-SY5Y cells and H9c2 cells were cultured in DMEM medium which were supplemented with 10% (*V*/*V*) fetal bovine serum (FBS) and 1% (*V*/*V*) penicillin-streptomycin (Thermo Technologies, New York, NY, USA). All of the cells were incubated in a humidified atmosphere containing 5% CO_2_ at 37 °C. The positive drug (edaravone), diosgenin and diosgenin-amino acids derivatives DG-1 to 18 were dissolved in DMSO (Sigma, St. Louis, MO, USA) and added at suitable concentrations to the cell culture.

#### 4.3.2. Neuroprotective Activity Test Using SH-SY5Y Cell

Analysis of neuroprotective effects was performed on SH-SY5Y cells via the MTT method. The SH-SY5Y cells in the logarithmic phase were plated into 96-well sterile plates at a density of 4.0 × 10^3^ cells/well and cultured at 37 °C with 5% CO_2_ containing incubator for 24 h. The tested compounds (0.1 mL) were added to the treatment group cells in final concentrations ranging from 3.125 to 50 μM, while the TBHP treated cells and the control-differentiated cells were treated with the same amount of medium (DMEM medium which were supplemented with 10% (*V*/*V*) fetal bovine serum (FBS) and 1% (*V*/*V*) penicillin-streptomycin). The following day, the well of treatment group and TBHP treated cells were cultured with 350 μM TBHP (0.1 mL) and incubated overnight. The control-differentiated cells were injected with new growth media at equal amounts. The cell culture fluid was replaced by 0.1 mL fresh medium before 20 µL MTT solution (5 mg/mL) was added to each well. After 4 h, the cell supernatant medium was removed by aspiration, then each well was given 0.15 mL DMSO and the plate was shaken 5 min. The optical density (OD) for each well was read using a BIORAD 550 spectrophotometer (Bio-Rad Life Science Development Ltd., Beijing, China) plate reader at 490 nm. All the results of damaged SH-SY5Y cells were carried out three times using replicate wells for each treatment. All the results of damaged SH-SY5Y cells were carried out three times using replicate wells for each treatment. The viability rates can be obtained from the following formula (1):
[OD_490_ (Treatment group) − OD_490_ (TBHP damaged cells)]/[OD_490_ (Control group) − OD_490_ (TBHP damaged cells)] × 100%(1)

The EC_50_ values were identified as the concentration of compound that could cause 50% of the individual effective drug concentration. The calculation formula (2) is as follows.
−pEC_50_ = logC_max_ − log 2 × (ΣP − 0.75 + 0.25P_max_ + 0.25P_min_)(2)
where C_max_ is the maximum concentration, ΣP is the sum of viability rates, P_max_ is the maximum value of the viability rate and P_min_ is the minimum value of the viability rate.

#### 4.3.3. Biosafety Evaluation Using H9c2 Cell

The H9c2 cells were transferred into individual wells of 96-well plate (4.0 × 10^3^ cells/well) and cultured 24 h at 37 °C with 5% CO_2_. Compounds DG-13 and DG-15 dissolved in DMSO were added to each well with required concentrations (3.125, 6.25, 12.5, 25, 50 µM). Then the control group were treated with the same volume of medium. The non-administered group were not treated (no cells and no medium). Then the sterile plates were further incubated for 4 h, following MTT addition. After 4 h, supernatant liquor was removed into the waste and all of the wells were then placed into the fresh DMSO (0.15 mL). The absorbance was measured using BIORAD 550 spectrophotometer with the emission wavelengths of 490 nm. Each data was measured three times in parallel survival fraction of cells, calculated by the equation: [OD_490_ (Drug group) − OD_490_ (non-administered group)]/[OD_490_ (Control group) − OD_490_ (non-administered group)] ×100%.

#### 4.3.4. Morphological Analysis Using Giemsa and DAPI Staining

The SH-SY5Y cells were seeded in 12-well sterile plates at a density of 3.0 × 10^4^ cells per well and placed at 37 °C in a 5% CO_2_ containing incubator for 24 h. Afterwards, the medium containing compound DG-15 were added to the cultures for 24 h and the concentration range of the examined compounds was 10, 20 and 40 µM. Then 0.1 mL TBHP was used to damage cells and the plate was cultured for further 24 h. Removing the culture medium, washing twice with PBS and fixing cells with 2 mL methanol for 2 min and 1 mL paraformaldehyde for 15 min respectively and washed with fresh PBS again. Giemsa solution or DAPI was added to stain the fixed cells for 2 min and protected from light. Finally, the cells ware washed with water and were observed under inverted fluorescent microscope.

#### 4.3.5. Neuroprotective Activity Test Using CAM model

The CAM was examined as a predictive model for the angiogenic activity studies of compounds DG-15. The wide end of the fresh fertilized egg was placed upward in an incubator at 37 °C with a relative humidity of 60%. All the eggs were turned every two hours to prevent from sticking to each other. Compound DG-15 was dissolved with acetone at a concentration of 1 mg/mL and 4 mg/mL and then 5 µL solution was transferred to gelatin sponge using a pipette respectively. The sham treatment group was added to the same volume of saline. Disinfected with UV lamp for 30 min. After 7 days, the broader said of eggs was disinfected with 70% alcohol, then a 30-mm-diameter window was carefully opened in the shell to reduce the chamber pressure before removed the upper shell membrane with tweezers. In addition, the non-fertilized oocytes and unhealthy developed embryos were thrown away. Then the gelatin sponge was embedded into the allantoic membrane and the window was sealed with adhesive tape, with the eggs being returned to the incubator. The eggs were then incubated 3 days after administration, fixed with methanol-acetone (1:1) for 10 min and repeated four times. In the end, the allantoic membrane was cut out, placed in deionized water and tiled on a glass slide. The vascular growth was analyzed with a dissecting microscope and counted by software of Image Professional Plus 5.0 software (Media Cybernetics, Inc, MD, USA)

### 4.4. Statistical Analysis

All experimental data were expressed as the means ± standard deviation (SD) of three replicates. The statistical analysis was performed by the software of SPSS (Version 20.0, International Business Machines Corp. New York, NY, USA) to calculate the variance. One-way analysis of variance (ANOVA) was performed to determine the significance between groups. A *p*-value of less than 0.05 was deemed to be significant.

## 5. Conclusions

In summary, eighteen diosgenin-amino acid derivatives were designed and synthesized by esterification reaction and deprotection reaction, respectively. The structures of all compounds were verified by ^1^H-NMR, ^13^C-NMR, HRMS. Then the MTT method was used for investigating the neuroprotective activity on the damaged SH-SY5Y cells and biological safety on H9c2 cells. The result indicated that DG-15 has a stronger neuroprotective activity and lower cytotoxicity than diosgenin. Interestingly, diosgenin attached to different types of amino acids, with and without protective groups, has different neuroprotective activities. And further Giemsa and DAPI staining explains the effect of compounds DG-15 on SH-SY5Y from the perspective of cell morphology. We also conducted a preliminary evaluation of the angiogenesis effect of the compounds DG-15 using the CAM model. The result proved that it could promote the formation of tiny blood vessels in chicken embryos damaged by TBHP. We conceive that this study could promote the attempt to discover more efficient and double-effective leading compounds from natural products.

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
