# Peer review of "Design, Synthesis and Biological Evaluation of Diosgenin-Amino Acid Derivatives with Dual Functions of Neuroprotection and Angiogenesis"

_molecules, 2019, doi:10.3390/molecules24224025_

Round 1

Reviewer 1 Report

In this manuscript, Li et al synthesize and characterize several Diosgenin – amino acid derivatives with neuroprotective and pro-angiogenic properties to be proposed as drug candidates for the treatment of ischemic stroke.  The topic is of high interest and surely deserves consideration. The paper is well written and clearly organized and the experiments are described in sufficient detail. The experimental results consistently support the main findings of the paper. However, I have some concerns that should be addressed before the acceptance of the paper. First, it is not clear to me why the authors decided to link diosgenin with amino-acids? what properties should they confer to the parent molecule? Second, I do not understand why there is not the derivative with Histidine.

Indeed, His may react with metal ions (as Zn and Cu) that may have a very relevant role in neuroprotection. It is known that copper ions and zinc may have an impact on proteasome functions and polyubiquitination reactions (see e.g Coordination Chemistry Reviews 347, 1-22, 2017; Chemistry-A European Journal 17 (41), 11596-11603; Inorganic chemistry 52 (16), 9567-9573, 2013). We think that this issue would need to be briefly discussed in the introduction to ensure complete coverage of the field.

Author Response

Dear reviewer:

  We would like to express our gratitude for your comments concerning our manuscript entitled “Design, Synthesis and Biological Evaluation of Diosgenin-Amino acid Derivatives with dual functions of neuroprotection and angiogenesis” (Manuscript NO. molecules-603890). And the manuscript has been revised in term of your comments; documents answering each question of you point by point were enclosed. All of us would like to know if there are something needed to be amended. And we wish it to be suitable for publication in international journal of molecular sciences.

Sincerely yours,

Haimin Lei

Reviewer 2 Report

Specific comments concerning the manuscript entitled “Design, Synthesis and Biological Evaluation of Diosgenin-Amino acid Derivatives with dual functions of neuroprotection and angiogenesis” under reference Molecules-603890.

The manuscript is rather well constructed and documented by many appropriate data. However, there are some points that deserve to be improved or checked before it can be accepted for publication in “Molecules” concerning the analytical aspect of the article.

1)   Please, remove the sentence “All the conjugates were characterized using 1H-NMR, 13C-NMR, and HRMS analyses” from the abstract. There is no relevant information in this sentence, as this article is not dedicated to “analytical chemistry”. Moreover, even if the synthesized molecules were characterized by NMR and HRMS, these results are absolutely not commented in the manuscript.

2)   Table 2: The confidence of data expressed as mean ± SD is only 68.2% which in my opinion too low. The confidence should reach 95% which means for n = 3, mean ± 1.837 SD using a t-distribution with 2 degrees of freedom (t = 3.182, √n = 1.732 and t/√n = 1.837). In contrast, the confidence of data expressed as mean ± SD with n = 6 is 95%. Consequently, I suggest to the authors to correct their interval of confidence or increase the number “n”.

3)   Figure 1: The error bars are not indicated.

4)   Line 218: I am really interested by the assignation of the following proton: 6.22 (t, J = 5.5 Hz, 1H). Regarding the other non aromatic Diosgenin-Amino acid derivatives, the first proton is found at δ (ppm) more or less 5.37 (brs, 1H). I suggest discussing the value of this high chemical shift.

5)   Reference section: journal names are not abbreviated.

Other remarks

6)   Line 39 to 40: remove the capital letter from Tissue, Heparin, Urokinase, Edaravone and Nimodipine.

7)   Line 212 : please correct "raection"

Author Response

(The authors gave the same response as above.)

Reviewer 3 Report

Cai et al. have studied the effect of diosgenin derivatives with dual functions on neuroprotection and angiogenesis. The new synthetic analogues of diosgenin had introduced different amino acids or BOC-protected amino acids, and their neuroprotective effects on SH-SY5Y cells in culture, toxicity on H9c2 cells and, finally, its possible angiogenic activity in the chorioallantoic egg model were investigated. The authors claim that the new molecules exert superior effects when compared with the parent compound.

The study is novel, the methods although appropriate, need to be extended to be understandable and more details should be given. At the same time, the results should be much better described and in many instances those results are overestimated or incorrect.

Major comments

Results

In the neuroprotective activity description (line 99-102) it should be explained that and oxidative stress protocol was used. Indeed toxicity was induced by addition of TBHP (that should be first written in full 2,4,5-Trihydroxybutyrophenone). Also add to the text that SH-SY5Y cells are human neuroblastoma cells.

A major problem is Table 2, the results therein and the methods that were applied.

The method (line 300 and onwards) rather than proliferation appears to assess viability by the MTT assay. In the case that actual proliferation of the cells was measured the control (no treatment or vehicle addition to the wells) should be shown and at different time periods, nd no toxic species added. The medium used after plating the cells is not stated (just DMEM with no serum?). Lines 393-394,  please modify the sentence: the cells are not added to the treatment, it is the reverse, treatment is added to the previously seeded cells. Instead of μl in the text the volumes are in ml.  “… decanted by aspiration…” should be “removed by aspiration”. And finally,  a complicated calculation formula was used, instead of a concentration-response curve and its calculations.

Results in Table 2.

Proliferation rate? It should be viability.

-It is at odd that many of the compounds show and EC50 >40 µM, even though the maximal concentration was 50 µM. Please change it.

-All the data should be double-checked. For instance, DG-8 or DG-9 appears to exert effects with a bell shape, how the authors decide which is the ED50? DG-18 has all negative results, but there is no comment in the text. Obviously this analogue is really toxic to the cells but the concentration-dependence if reversed (higher toxicity at lower concentrations).

Boisafety evaluation

In the text is mentioned (line 122) that “…diosgenin itself has certain toxicity…” when at the lowest concentration tested the reduction in viability is more that 60%!!!. Please rewrite the sentence explaining that diosgenin and DG-13 are toxic, but DG-15 was less toxic and as stated in the text the survival of the cells in its presence was improved.

Promotes Angiogenesis.

The density or number of blood vessels can be assessed by different ways. In this manuscript, again, the method used is not clear. One can measure the area covered by vessels (actual area, or percentage of total area). The number of vessels could be counted (with a grid and counting grid crossings), etc… Just looking the results of CAM in Fig. 4 one can see that in the control situation there are vessels of different sizes (the segmentation taking into the account the size should be stated) and that bigger (primary) branches have secondary and tertiary branches, this morphology is maintained after 1 mg/ml DG-15 treatment. However, after 4 mg/ml treatment the vessels adopt a radial morphology and they bear few if any branches (secondary and tertiary). This should be described in the text. Moreover, the assessment of the vessels should be better performed because those micrographs do not show and increased angiogenesis at all.

Blank (line 54) should be changed to no treatment, vehicle treatment or sham treatment (sponge with no drug). Add the time (3 days) of treatment in the text.

Discussion does not exist. It is just a summary of the results or Conclusions. The results of the chemical synthesis and that of the biological activity should be discussed in light of the previous literature.

Minor comments

Abstract Page 1. “…as a vital component of steroidal compound, …” It is not clear what does this mean. Change to “…a natural product with steroidal structure,…”

Gimesa (line 25), likely a typographical error, first appear in the Abstract. Please check all the text and correct to Giemsa.

Line 28. “this founding…” change to finding.

-Page 2, line 44, pro-clinical should be pre-clinical

-Line 57, “…have been carried out that…” should read “…have shown that…”

-Line 133, appearation change for appearance

-Line 137, formated should be formation

-Line 140, lowed should be reduced

-Fig. 2 legend. Reword the whole legend. Gimesa change by Giemsa, “induced by” by “treated with”. Model group: were those cells treated with TBHP? Make that clear. The magnification of the micrographs is not 200x, state that the bar in the picture indicates 200 μm.

-Line 395, change model group by TBHP treated cells.

-DMSO is toxic to the cells, how much was the highest % of DMSO in the wells? Did all the concentrations of drugs have the same percentage of DMSO?

-Absorbance does not have excitation wavelength, just emission.

-Lines 410-411 reword the sentence, cells were incubated for 4h and not 24h after MTT addition.

-Avoid using blank group, model group

Author Response

(The authors gave the same response as above.)

Round 2

Reviewer 3 Report

The MS has been improved and many of the changes required by reviewer 1 had been taken into account.

However, the authors do not understand the comments and what is requested in the revised version of the work. It would be desirable that an English speaking person translates the requests of the reviewer.

The amendments in the MS are in red to highlight them. Please in the answer to the reviewer do not include what has been changed, only state the page or lines corrected.

-The molecules’ structures in Table 1 had been deleted? If so, why? They were informative.

-Lines 147-149. Although not requested, a sentence about the toxicity model used (oxidative stress) has been added to revised version. Keep that change since it is appropriate.

-In Table 2, proliferation rate has been not changed to viability. This reviewer insists that the authors did not assess proliferation but viability in the cell cultures. Increased viability under a given treatment could be due to increased proliferation or increased survival, which should be investigated in further experiments. Change “proliferation” to “viability” as requested.

Delete proliferation and write viability, not just in the Fig. also in the text, including the text of the heading.

“Compound Proliferation rate (%) EC50”    change to “Compound Viability rate (%) EC50”

-Line 174. Diosgenin was toxic, not little toxic. Delete little.

-Line 233. Legend to Figs 2 and 3. The authors have not corrected what I asked to change. Please write just this: “Figure 2. Giemsa staining on SH-SY5Y cells treated with compound DG-15 with different concentrations: (a) Control group; (b) TBHP treated cells; (c) 10 μM; (d) 20 μM; (e) 40 μM. The cell morphology was observed under the fluorescence microscope. The most representative fields were shown. Calibration bar: 200 μm.” The same change should be done in the legend to Fig. 3.

-Promotion of angiogenesis. Lines 242-246. Now what has been done is better explained.

-Discussion has been revised and improved.

-Neuroprotective Activity Test Using SH-SY5Y Cell. This section has been modified.

The calculation formula of the EC50 now is better explained. Nevertheless, it is the first time that I see its being used, in the era of software used for that kind of calculations.

-Finally, DMSO is toxic and its actual concentration is really important.

The authors have incorrectly worked out the concentration of DMSO: they diluted 10 µL in 1000 μl

Using this equation V x %= V’ x %’    10 μl x 100%= 1000 μl x ?%

?%= 10 x 100 %/ 1000;   ? %= 1%

Author Response

Dear reviewer:

  We appreciate very much your patient comment and guidance concerning our manuscript entitled “Design, Synthesis and Biological Evaluation of Diosgenin-Amino acid Derivatives with dual functions of neuroprotection and angiogenesis” (Manuscript NO. molecules-603890). We've made appropriate modifications according to the advice point by point. All of us would like to know if there are something needed to be further improved. And we sincerely wish it to be suitable for publication in international journal of molecular sciences.

Sincerely yours,

Haimin Lei
